# Long-Term Efficacy and Safety of Evinacumab in Patients with Homozygous Familial Hypercholesterolemia: Real-World Clinical Experience

**DOI:** 10.3390/ph15111389

**Published:** 2022-11-11

**Authors:** Claudia Stefanutti, Dick C. Chan, Serafina Di Giacomo, Claudia Morozzi, Gerald F. Watts

**Affiliations:** 1Department of Molecular Medicine, Lipid Clinic and Atherosclerosis Prevention Centre, ‘Umberto I’ Hospital ‘Sapienza’ University of Rome, I-00161 Rome, Italy; 2Medicine School, University of Western Australia, Perth, WA 6000, Australia; 3Lipid Disorders Clinic, Department of Cardiology and Internal Medicine, Royal Perth Hospital, Perth, WA 6000, Australia

**Keywords:** angiopoietin-like 3 protein inhibitors, atherosclerotic cardiovascular disease, familial hypercholesterolemia, lipoprotein apheresis, low-density lipoprotein

## Abstract

Homozygous familial hypercholesterolemia (HoFH) is a rare, genetic condition characterized by markedly elevated plasma low-density lipoprotein cholesterol (LDL-C) concentrations from birth and increased risk of premature atherosclerotic cardiovascular disease. Evinacumab is an inhibitor of angiopoietin-like 3 protein that offers a new approach for correcting high LDL-C in HoFH. Evinacumab was administered intravenously (15 mg/kg Q4W) for 24 months in 7 patients with genetically confirmed HoFH, receiving background lipoprotein apheresis (LA) and/or lipid-lowering treatment (LLT). Assessment of efficacy and safety were carried out before and after 24 months of evinacumab treatment. The LDL-C lowering effect of evinacumab without LA were also investigated in the 7 HoFH patients after a subsequent compassionate extension period. Twenty-four months of treatment with evinacumab against background LA and LLT resulted in a significant reduction in LDL-C (−46.8%; *p* < 0.001). LDL-C reduction with evinacumab was maintained during the compassionate extensions period in the absence of treatment with LA (−43.4%; mean follow-up of 208 ± 90 days). Evinacumab was well-tolerated, with no major adverse event reported or significant changes in liver and muscle enzyme concentrations. Our findings suggest that evinacumab is a safe and effective treatment for patients with HoFH receiving best standard of care in a routine setting.

## 1. Introduction

Homozygous familial hypercholesterolemia (HoFH) is a rare, life-threatening, genetic disorder with a prevalence of 1:300,000–1:360,000 [1]. The disorder is characterized by biallelic mutations in genes regulating activity of the low-density lipoprotein (LDL) receptor [2], leading to markedly elevated plasma LDL-cholesterol (LDL-C) concentrations from birth and increased risk of premature atherosclerotic cardiovascular disease (ASCVD) [1,3,4,5]. Despite intensive lipid-lowering treatment (LLT), a significant proportion of HoFH patients remain at high risk of ASCVD, owing to not achieving LDL-C goals [1,6]. Hence, there is an unmet need for effective treatments for patients with HoFH [1,7,8].

Lipoprotein apheresis (LA) is highly effective tool in lowering LDL-C in HoFH patients [1,8,9]. A combination of LA with lipid-lowering drugs is recommended to achieve further lowering of LDL-C levels, and limit the rebound of LDL-C. However, current lipid-lowering drugs, such as statins, ezetimibe and proprotein convertase subtilisin/kexin type 9 (PCSK9) inhibitors, are ineffective in reducing LDL-C in HoFH to guideline-recommended LDL-C goals [1]. This is because of a substantial reduction in or virtually complete absence of LDL receptor expression in the true genetic bi-allelic defect of HoFH [2]. New therapies are needed to target LDL metabolism beyond the LDL receptor pathway in patients with HoFH, to be used alone or in combination with LA [1,7,8].

Angiopoietin-like 3 protein (ANGPTL3) is a hepatokine that is exclusively secreted by the liver [10]. It plays a key physiological role in the regulation of plasma lipid and lipoproteins’ metabolism that involves the inhibition of lipoprotein (LPL) and endothelial lipases (EL) [10,11]. Inhibition of ANGPTL3 can offer a new approach for correcting dyslipidemia, including FH [7,12,13]. Evinacumab is a fully human monoclonal antibody that specifically inhibits ANGPTL3 extracellularly [13,14]. While the precise mechanistic action of evinacumab on LDL metabolism remains uncertain [12,15,16], several recent clinical trials have demonstrated that evinacumab can effectively lower plasma LDL-C levels in patients with HoFH with a safe and well-tolerated profile [14,17,18,19]. Although evinacumab appears efficacious in lowering LDL-C in clinical trials, its long-term efficacy and safety have not yet been formally evaluated in real-world clinical practice in HoFH patients.

In the present study, we assessed the long-term efficacy and safety of evinacumab in HoFH patients on and off LA in real-world clinical practice. The primary outcome was the percentage changes in the calculated LDL-C from baseline to 24 months after evinacumab and LA treatment. The effect of evinacumab on other lipid, lipoprotein and apolipoprotein (apo) levels, including apoC-III and lipoprotein (a) (Lp(a)), was also investigated.

## 2. Results

### 2.1. Patient Characteristics

Table 1 shows the demographic, clinical, genetic and treatment history of the seven HoFH patients studied (five females and two males; mean age 43 ± 16 years). Two patients were diagnosed with true HoFH and five with compound heterozygous FH (HeFH). More than half had hypertension, but none had type 2 diabetes and smoking habits. A history of coronary artery disease (CAD) and/or aortic valve disease (AVD) was found in four patients. Prior to the administration of the first dose of evinacumab, six patients received treatment with LA, either weekly (*n* = 2) or biweekly (*n* = 4), and a range of lipid-lowering therapies, including rosuvastatin (*n* = 4), simvastatin (*n* = 2), ezetimibe (*n* = 7), evolocumab (*n* = 4), alirocumab (*n* = 1) and lomitapide (*n* = 1). The mean levels of total cholesterol (TC), triglyceride (TG), high-density lipoprotein-cholesterol (HDL-C) and LDL-C were 9.1 ± 1.6 mmol/L, 1.2 ± 0.51 mmol/L, 1.1 ± 0.29 mmol/L and 7.4 ± 1.7 mmol/L, respectively. Only one patient had never undergone LA, either before or during the study, being treated only with rosuvastatin and ezetimibe.

### 2.2. Effect of Evinacumab and Lipoprotein Apheresis

As seen in Figure 1A, evinacumab with LA treatment showed substantial and consistent LDL-C reduction relative to baseline across all follow-up visits after baseline in all individual HoFH patients, with the exception of Patient 1; in this patient, the corresponding on-treatment plasma concentration of LDL-C at 24-month follow-up returned approximately to pre-treatment levels, but improved further during the compassionate use extension phase. The patient discontinued LLT for hospitalization due to a condition unrelated to her severe hypercholesterolemia; LLT was re-instituted following recovery, with subsequently marked and sustained reduction in LDL-C. Compared with baseline, the mean LDL-C reductions were −54.4%, −48.9%, −49.4% and −46.8%, respectively, at 6, 12, 18 and 24 months after evinacumab and LA treatment (Figure 1B; *p* < 0.001 for all compared with baseline). The percentage change from baseline in LDL-C at 24-month follow-up for individual patients is shown in Figure 2. As seen, a similar mean reduction in LDL-C was found in the compassionate evinacumab use extension phase without background LA treatment (−43.4%; mean follow-up of 208 ± 90 days).

Overall, the LDL-C lowering effect of evinacumab with or without LA treatment was greater than LA alone (i.e., without evinacumab treatment). In the latter, time-average LDL-C was reduced by 27.2% in the six patients who received LA during the normal course of their therapy prior to evinacumab (Appendix A. Among the seven patients treated with evinacumab and LA, a reduction in the LDL-C level of ≥30% was observed in six patients; four patients achieved an LDL-C reduction ≥ 50% with two patients having on-treatment LDL-C level < 2.5 mmol/L (97 mg/dL). Likewise, evinacumab with LA treatment also significantly reduced plasma concentration of TC (−44.5%), non-HDL-cholesterol (−46.6%) and apoB (−33.8%) at 24-month follow-up (*p* < 0.001 for all) in a manner similar to that of LDL-C (Figure 3 and Figure 4). Individual responses in plasma concentrations of TC, non-HDL-C and apoB during 24-month treatment are shown in Appendix A. As expected, the percentage reduction in LDL-C concentration from baseline to 24-month follow-up was significantly correlated (*p* < 0.001 in all) with the corresponding reductions in TC (*r* = 0.996), non-HDL-cholesterol (*r* = 0.999) and apoB (*r* = 0.963).

Reductions in plasma levels of TG (−46.6%); HDL-C (−29.9%); remnant-cholesterol (REM-C) (−46.5%); apoA-I (−26.0%); apoC-III (−72.8%) and Lp(a) (−32.2%) were also observed with evinacumab treatment at 24-month follow-up (Figure 5). There was no significant correlation between the percentage fall in LDL-C concentration and the corresponding changes in TG, REM-C, apoA-I, apoC-III and Lp(a) concentrations (data not shown). However, the effects of evinacumab on plasma hsCRP were variable among seven HoFH patients; plasma high-sensitivity C-reactive protein (hsCRP) concentration decreased in five patients (ranging from −9.1% to −52.9%) and increased in two patients (+55.6% and +75.0%, respectively) with an overall reduction of 8.9% (Appendix A). Full details of plasma lipid, lipoprotein and apolipoproteins concentrations in individual HoFH patients before and during therapy with evinacumab are shown in Appendix A.

### 2.3. Adverse Effects of Evinacumab Therapy

Evinacumab treatment was generally well-tolerated; none led to evinacumab discontinuation owing to severe adverse effects. There was no cardiovascular event observed during the 24-month follow-up and subsequent compassionate extension period (12 months) with evinacumab. Injection-site reactions were not reported. None of the patients reported any symptoms related to the more common adverse effects (pharyngitis, nasal congestion, myalgia, diarrhea and arthralgia) reported elsewhere. Overall, plasma aspartate transaminase (AST), alanine transaminase (ALT) and creatine kinase (CK) concentrations for individual HoFH patients remained stable with treatment with evinacumab, the corresponding liver and enzyme concentrations not exceeding three times the upper limit of normal (ULN) (Figure 6).

## 3. Discussion

Our principal finding was that in patients with HoFH receiving the best standard of care including lipoprotein apheresis, ANGPTL3 inhibition with evinacumab was effective in lowering the LDL-C concentration and was well tolerated over a period of 24 months. We further demonstrated a similar reduction in LDL-C with the compassionate use of evinacumab alone. Evinacumab also reduced plasma concentrations of TC, TG, HDL-C, non-HDL-C, REM-C, apoA-I, apoB, apoC-III, Lp(a) and, to a lesser extent, hsCRP.

### 3.1. Previous Studies

Few clinical trials have reported the efficacy and safety of evinacumab in lowering LDL-C concentration in patients with HoFH [17,18,19]. In a phase 2 study, involving nine patients with genetically confirmed HoFH, evinacumab (250 mg sc; 15 mg/kg at week 2) was associated with a mean decrease in LDL-C of 49% at week 4 [17]. In a large phase 3 trial of 65 HoFH patients (ELIPSE HoFH) with or without LA, a significant decrease in plasma LDL-C (−47%) was seen at week 24 and 48 weeks (−47% and −46%, respectively) in those receiving evinacumab (15 mg/kg IV Q4W) with no serious adverse events of active treatment reported [18,19]. However, the diagnosis of HoFH in this study was not exclusively based on genetic criteria [18,19]. More importantly, these studies were not completed in a real-world clinical setting. In a case report of a patient with genetically confirmed HoFH receiving a statin, ezetimibe and evolocumab, a 17-month treatment with evinacumab (15 mg/kg IV Q4W) was associated with marked reduction in LDL-C levels (−37%) and decreased frequency of LA with no self-reported adverse effects [20]. In another study of two severely affected young HoFH patients, evinacumab reduced plasma LDL-C concentration by approximately 50% [21]. However, these two studies did not report the effect of evinacumab on other lipid and lipoprotein profiles. We have extended previous studies by investigating the effect of evinacumab on plasma LDL-C and other atherogenic lipid and lipoprotein concentrations in seven patients with genetically confirmed HoFH attending a clinic in Rome. We also compared the LDL-C lowering effect of evinacumab with and without LA treatment.

### 3.2. Mechanisms

Several mechanisms of the action of ANGPTL3 inhibition on triglyceride-rich lipoprotein (TRL), LDL and HDL metabolism have been postulated via the control of LPL and EL [13,15,16]. An increase in LPL activity with ANGPTL3 inhibition enhances the peripheral clearance of both chylomicron and very-low-density lipoprotein (VLDL) particles, contributing to the marked fall in plasma triglycerides and TRLs. Direct uptake by the liver of VLDL remnants, together with the fall in the hepatic output of VLDL may contribute to the reduction in the production of LDL particles and LDL-C concentration [15]. ANGPTL3 inhibition may also increase the activity of EL [15]. This effect enhances the hydrolysis of HDL-phospholipids and upregulates the catabolism of HDL particles, resulting in a reduction in the plasma HDL-C concentration. Accordingly, we found that evinacumab treatment resulted in substantial reductions in plasma concentrations of TC, TG, HDL-C, LDL-C, non-HDL-C, REM-C, apoA-I and apoB. The exact mechanism of LDL reduction with evinacumab remains uncertain [12,15,16]. The potent LDL-lowering effect of evinacumab appeared to be independent of the degree of LDL receptor activity [18], as observed in our HoFH patients who had minimal or no residual LDL receptor function. Consistent with this finding, the experimental data show that evinacumab had no effect on LDL-receptor functional activity in lymphocyte studies [22]. A stable isotope study in four patients has also suggested that evinacumab reduced VLDL-apoB production and increased LDL-apoB clearance in patients with HoFH via an LDL receptor-independent pathways mediated by EL [16]. This suggest that evinacumab may enhance the clearance of LDL via other receptors or unidentified non-receptor-mediated mechanism. However, this speculation remains to be investigated. The reduction in apoC-III concentration may be a consequence of reduced secretion and enhanced clearance of apoB-containing lipoproteins with evinacumab. The mechanistic effect on lowering plasma Lp(a) concentration is unclear but may be due to a reduction in apoB availability for coupling apo(a) to form Lp(a) [23]. It is noteworthy that evinacumab showed no significant effect on plasma Lp(a) concentration in clinical trials [17,18].

### 3.3. Strengths and Limitations

This study was conducted in patients with genetically confirmed HoFH. It provided a long-term follow-up period of 24 months to assess the efficacy and safety profile of evinacumab in real-world clinical practice. We also compared the effect of evinacumab with and without background LA on plasma LDL-C concentration.

Our study does have limitations. The sample size was relatively small, but the lowering effect of LDL-C were comparable to previous clinical trials of evinacumab [17,18,19]. However, the sample size was not sufficient to allow for subgroup analysis on gender and/or type of mutation (true FH vs. compound HeFH). We did not employ a formal placebo-controlled design to study the effects of evinacumab treatment on plasma LDL-C concentrations because of pragmatic reasons. We did not study effects on inflammation and vascular outcomes, but one study has shown profound reduction in atherosclerotic plaque progression with the use of evinacumab in two HoFH patients [21]. Carotid intima-media thickness (IMT) has also been used as a surrogate marker for ASCVD in FH patients. Whether evinacumab therapy has a favorable effect on the progression of carotid IMT is under investigation in our HoFH patients.

### 3.4. Clinical Implication

Patients with HoFH are at extremely high-risk of ASCVD due to markedly elevations in LDL-C present from birth [1,3,4,5,6]. If left untreated, they have an extremely high risk of premature ASCVD and death in childhood or the teenage period [3,4]. Despite major advances, lipid lowering strategies remain very challenging in patients with HoFH [1,8]. Lomitapide is approved by the European Medicines Agency (EMA) for clinical use in Italy. Currently, four of our FH patients are subjected to lomitapide with variable doses, but not exceeding 30 mg /day. Only one patient of the ‘evinacumab’ sub-cohort continued lomitapide treatment. In our experience, evinacumab is superior to lomitapide with less side effects deriving from the dosage increase in lomitapide. LA is an effective treatment in HoFH patients [24,25], particularly when concomitantly employed with conventional lipid-lowering drugs. However, treatment responses to statins and PCSK9 inhibitors in HoFH patients are suboptimal because these agents require residual LDL receptor function [1,8]. Given that the mechanism of action of ANGPTL3 inhibitors appears to be independent of LDL receptor activity [18,22], evinacumab constitutes a major advance in therapy for managing patients with HoFH who have minimal or no residual LDL receptor function [12,14]. Accordingly, evinacumab has been approved by the US Food and Drug Administration (FDA) and, more recently, the European Medicines Agency (EMA) for the primary indication of HoFH [26,27].

In the present study, we demonstrated that treatment with evinacumab can effectively lower the plasma concentrations of LDL-C and other atherogenic lipoproteins in patients with HoFH receiving guideline-recommended combination therapy with or without LA. This suggest that evinacumab may provide a realistic therapeutic option for achieving acceptable LDL-C goals in HoFH without the need for time-consuming LA process, unless the reduction in atherogenic lipoproteins achieved with evinacumab is insufficient, or the HoFH patient has a concomitant increase in plasma LDL and Lp(a) levels. This condition strongly increases the individual cardiovascular risk and the increase in Lp(a), at the moment, can only be corrected by treatment with LA. Whether the substantial reductions in LDL-C and other atherogenic lipid and lipoprotein profiles with evinacumab and other ANGPTL3 inhibitors translates into a decrease in ASCVD events or improvement in vascular function and/or arterial inflammation in HoFH remains to be shown.

## 4. Materials and Methods

### 4.1. Patients and Study Design

Seven patients (5 females and 2 males) with HoFH from the Extracorporeal Therapeutic Techniques Unit, Rare Metabolic Diseases Centre, Umberto I Hospital, Sapienza University of Rome, Italy, were included in the study. Criteria for the diagnosis of HoFH were confirmed by the presence of 2 pathogenic mutant alleles at the *LDLR*, *APOB* or *PCSK9* loci and further defined as true HoFH (identical mutation in each allele of the same gene) and compound heterozygous FH (nonidentical mutations in each allele of the same gene) [1,2]. All patients received a wide range of lipid-lowering therapies, including LA (6 out of 7 using the Liposorber system MA-03; Kaneka Corp, Osaka, Japan), with adsorption columns containing negatively charged dextran sulfate (polyanion) bound on cellulose beads. All patients received an intravenous infusion of evinacumab at a dose of 15 mg per kg of body weight) every 4 weeks immediately after LA treatment and followed up in the normal course of their care. The assessment of efficacy and safety was carried out before and after 6, 12, 18 and 24 months of evinacumab treatment. All patients were subsequently transitioned into a compassionate program to continue on evinacumab at the same dose without LA treatment. The consort diagram for the study is shown in Figure 7.

### 4.2. Biochemical Analyses

Plasma AST, ALT, CK, TC, TG and HDL-C concentrations were determined by standard routine enzymatic methods. LDL-C concentrations were calculated with the Friedewald formula. Non-HDL-C was calculated by subtracting HDL-C from TC, whereas REM-C was calculated by subtracting HDL-C and LDL-C from TC. ApoA-I, apoB, apoC-III and hsCRP concentrations were determined by immunoassay. Lp(a) was measured by immunonephelometric assay (Behring Nephelometer II; Dade Behring Inc, Milton Keynes, UK.). The treatment effect of LA in 6 HoFH patients prior to evinacumab were expressed as time-average LDL-C using Kroon’s equation (Cmean = Cmin + K(Cmax − Cmin), where K is the rebound coefficient for HoFH [28,29].

### 4.3. Safety Monitoring

Adverse events were monitored during the course of their care. Adverse events during evinacumab treatment were defined as general systemic symptoms and injection site reactions. General systemic symptoms were flu-like (pharyngitis, nasal congestion), musculoskeletal (myalgia, back pain, arthralgia), gastrointestinal (diarrhea, constipation, nausea, abdominal discomfort/pain), and other (headache and fatigue). Injection site reactions were defined as a cluster of erythema, pain, bruising, swelling, induration, rash, or pruritus around the injection sites. Liver (AST and ALT) and muscle enzymes (CK) were measured at baseline and after 6, 12, 18 and 24 months of evinacumab treatment.

### 4.4. Statistical Analysis

Statistical analyses were carried out using the SPSS Statistics (Version 25; Armonk, New York; IBM Corp, access date: 10 July 2022.). Data were presented as mean ± SD unless otherwise specified. The primary outcome was the changes in the calculated LDL-C from baseline (i.e., the last calculated LDL-C value obtained before the administration of the first dose of evinacumab) to 24 months after evinacumab treatment. Treatment effects were analyzed using paired-*t*-test or Wilcoxon signed rank test where applicable. Associations between changes in LDL-C concentrations and other lipid and lipoprotein variables were assessed by simple linear regression. Statistical significance was defined at the 5% level.

## 5. Conclusions

Our findings suggest that evinacumab is a safe and effective treatment for patients with HoFH receiving best standard of care in a routine setting. 

## Figures and Tables

**Figure 1 pharmaceuticals-15-01389-f001:**
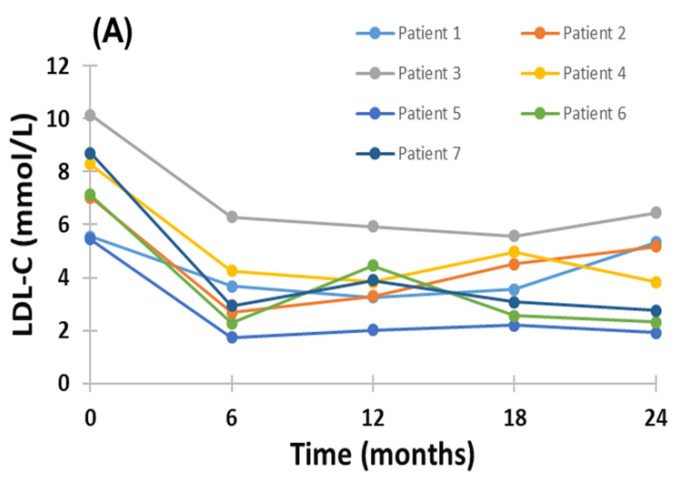
Change in plasma LDL-C concentration during 24-month treatment with evinacumab and LA in HoFH patients; individual (**A**) and mean (**B**) responses of patients. Mean ± SEM * *p* < 0.001 compared to baseline. No significant difference in plasma LDL-C concentration at 6, 12, 18 and 24 months.

**Figure 2 pharmaceuticals-15-01389-f002:**
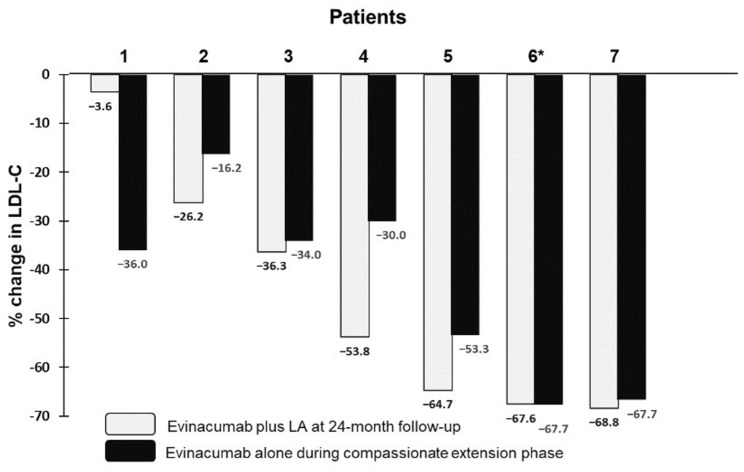
Percentage change from baseline in LDLC at 24-month follow-up (evinacumab plus LA) and during the compassionate extension phase follow-up (evinacumab alone) for individual patients. * Patient 6 received evinacumab treatment only during 24-month follow-up and the compassionate extension phase.

**Figure 3 pharmaceuticals-15-01389-f003:**
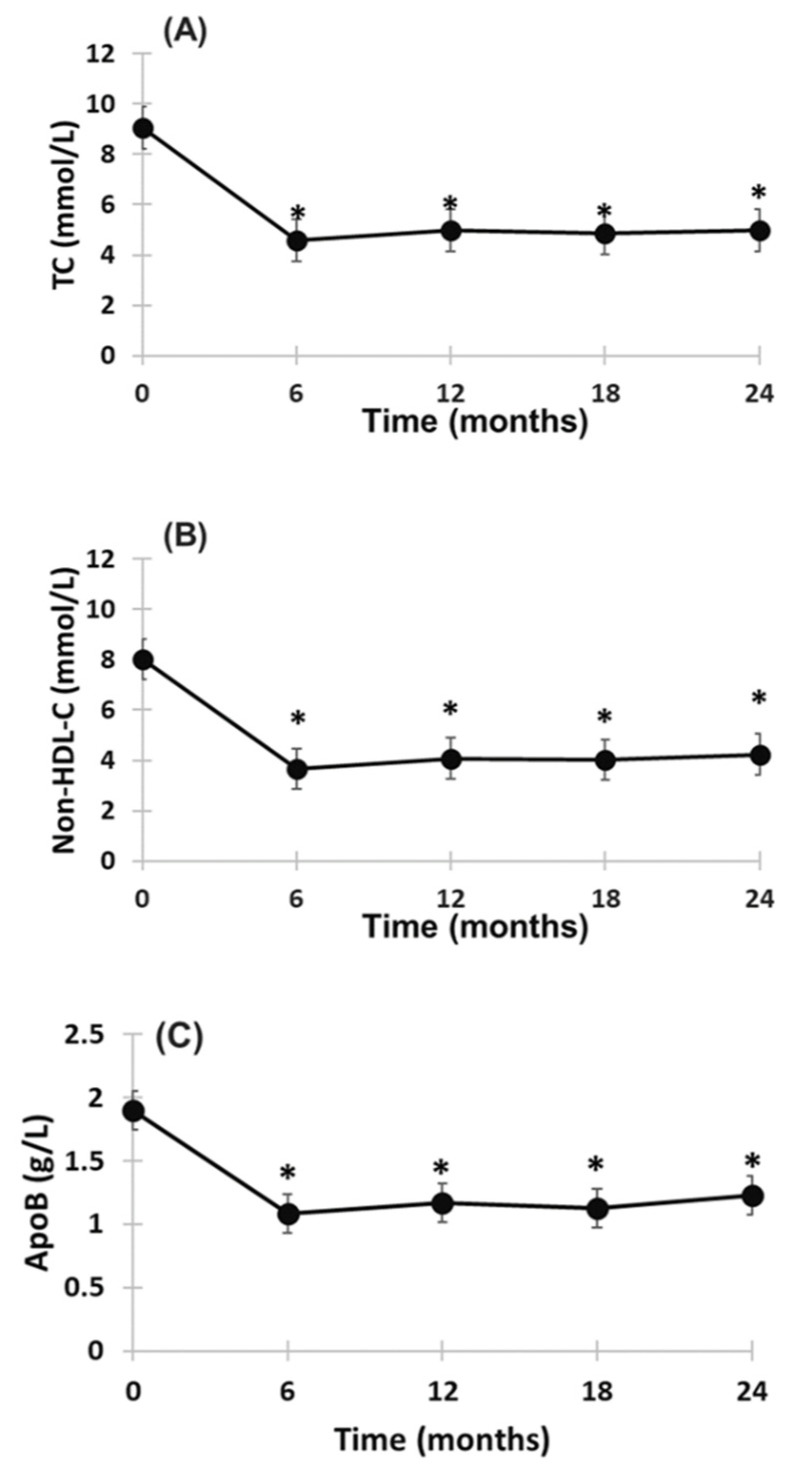
Change from baseline in TC (**A**), non-HDL-C (**B**) and apoB (**C**) during 24-month treatment with evinacumab and LA. Mean ± SEM. * *p* < 0.001 compared to baseline; No significant difference in plasma TC, non-HDL-C and apoB concentrations at 6, 12, 18 and 24 months.

**Figure 4 pharmaceuticals-15-01389-f004:**
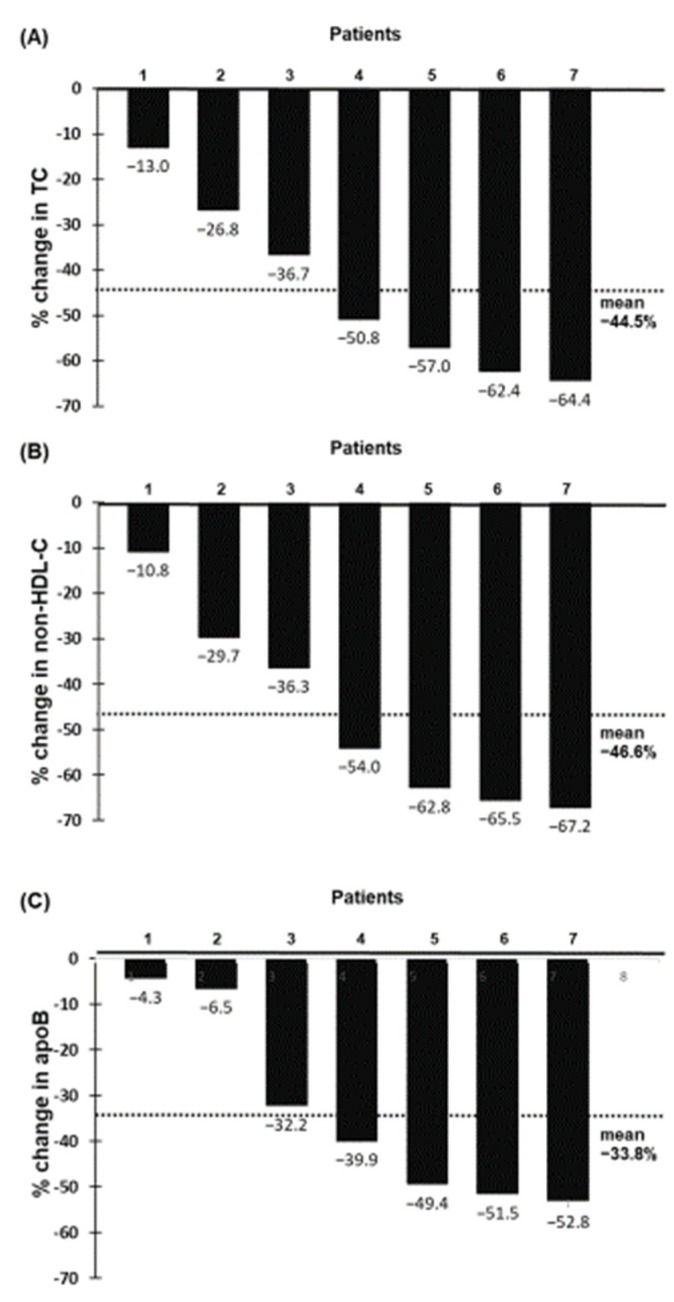
Percentage change from baseline in total cholesterol (**A**), non-HDL-cholesterol (**B**) and apoB (**C**) at 24-month follow-up for individual patients.

**Figure 5 pharmaceuticals-15-01389-f005:**
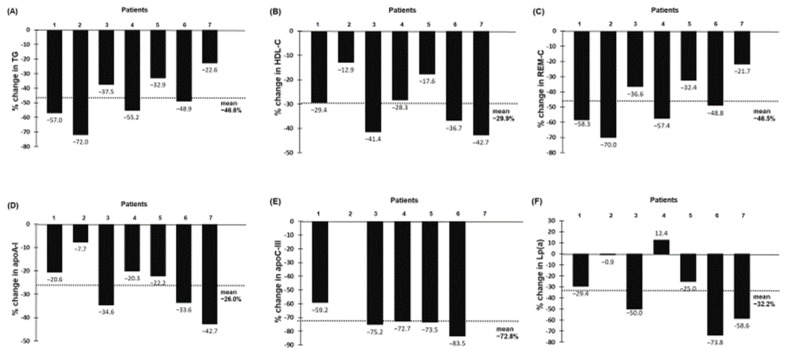
Percentage change from baseline in TG (**A**), HDL-C (**B**), REM-C (**C**), apoA-I (**D**), apoC-III (**E**) and Lp(a) (**F**) at 24-month follow-up for individual patient. Baseline apoC-III data were not available in 2 HoFH (patients 2 and 7).

**Figure 6 pharmaceuticals-15-01389-f006:**
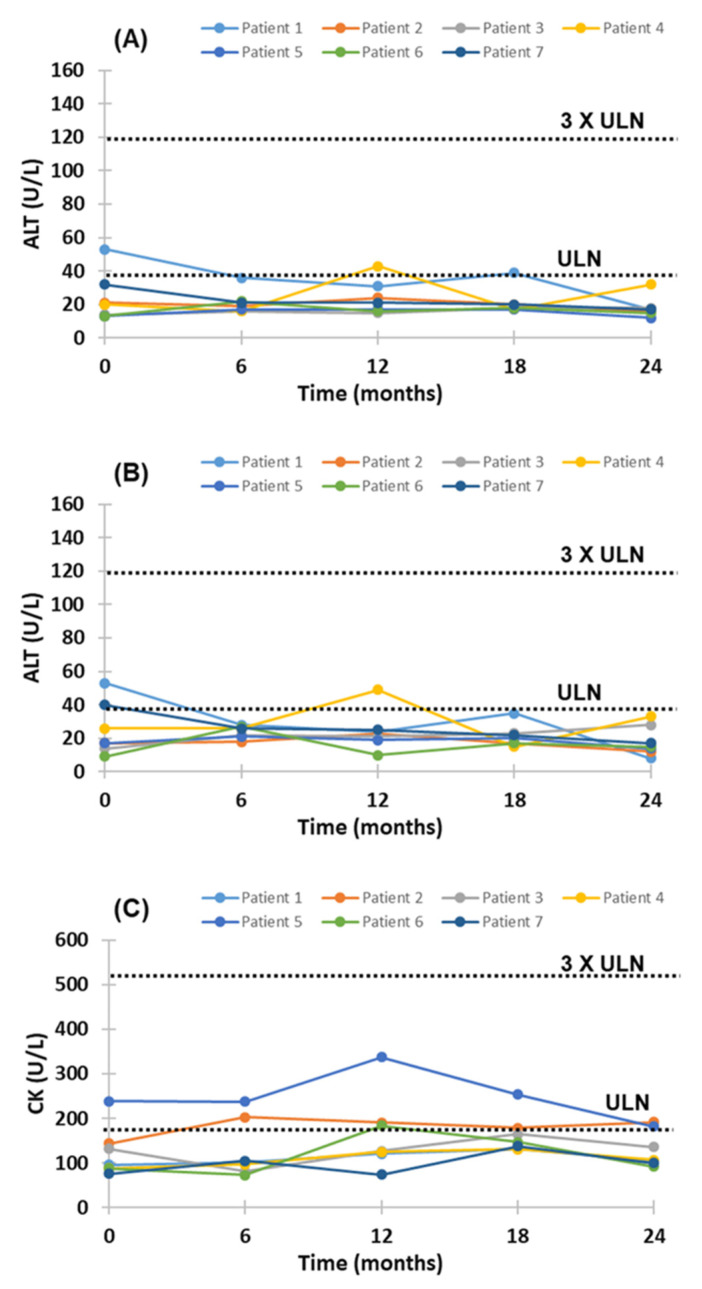
Change from baseline in plasma aspartate transaminase (**A**), alanine transaminase (**B**) and creatine kinase (**C**) during 24-month treatment with evinacumab. Dotted lines represent upper limit of normal (ULN) and 3X ULN for aspartate transaminase (AST), alanine transaminase (ALT) and creatine kinase (CK).

**Figure 7 pharmaceuticals-15-01389-f007:**
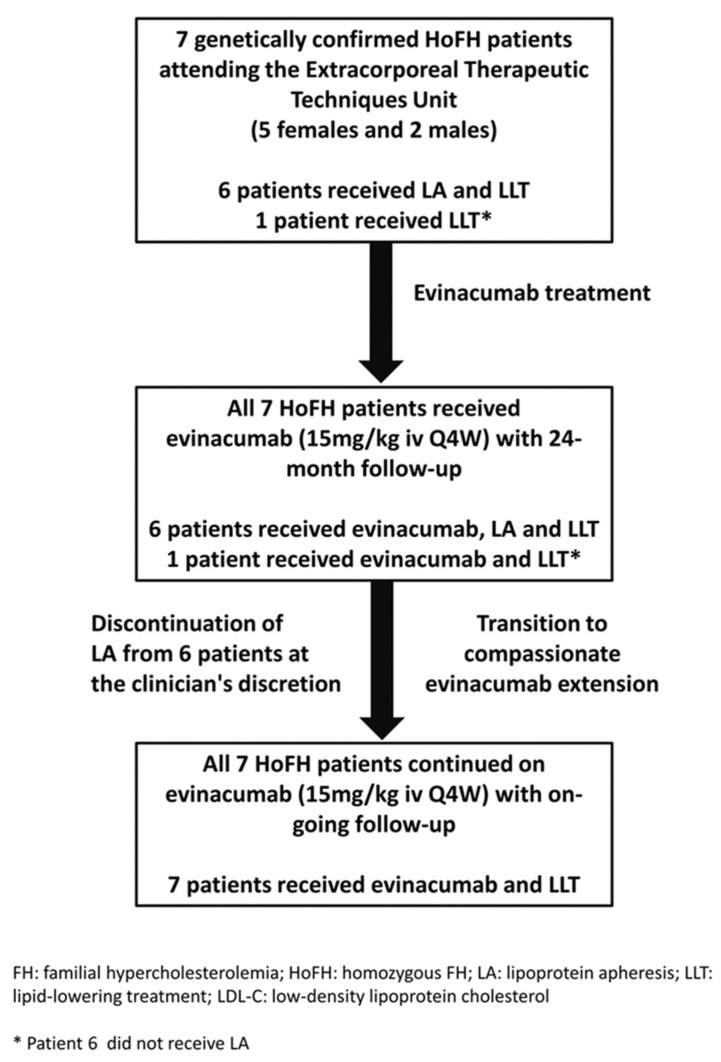
Consort diagram for the study.

**Table 1 pharmaceuticals-15-01389-t001:** Demographic, clinical, genetic and treatment characteristics of the patients with HoFH.

Patient	Age	Gender	FH Diagnosis	Mutation	Co-Morbidities	Lipid-Lowering
	(years)					Treatment
1	63	Female	Compound	c.1474 G > A, Asp471Asn	Premature CAD	Simvastatin 40 mg/day
			HeFH	c.2094 C > G, Cys677Trp	Hypertension	Ezetimibe 10 mg/day
						Alirocumab 150 mg/sc/15 days
2	36	Male	True HoFH	c.1056 C > G,		Ezetimibe 10 mg/day
				p.C352W(C331W) receptor		Lomitapide 20 mg/day
				defective		Evolocumab 140 mg/sc/15 days
3	14	Male	True HoFH	c.1478-1479 del 2bp (CT)	Premature CAD	Rosuvastatin 40 mg/day
				receptor negative		Ezetimibe 10 mg/day
4	52	Female	True HoFH	c.2054 C > T, Pro664Leu	Premature CAD	Simvastatin 40 mg/day
				receptor defective	Hypertension	Ezetimibe 10 mg/day
						Evolocumab 140 mg/sc/15 days
5	53	Female	True HoFH	c.2054 C > T,	Aortic valve disease	Rosuvastatin 20 mg/day
				p.P685L (P664L)	Hypertension	Ezetimibe 10 mg/day
						Evolocumab 140 mg/sc/15 days
6	50	Female	True HoFH	c.2054 C > T,	Hypertension	Rosuvastatin 40 mg/day
				p.P685L (P664L)		Ezetimibe 10 mg/day
7	34	Female	Compound	Deletion of promotor and		Rosuvastatin 20 mg/day
			HeFH	Ex 1-2, c.1775 G > A,		Ezetimibe 10 mg/day
				Gly571Glu		Evolocumab 420 mg/sc/30days

CAD: coronary artery disease; FH: familial hypercholesterolemia; HeFH: heterozygous FH; HoFH: homozygous FH. Lipoprotein Apheresis: Patients 1 and 4 (weekly); Patients 2, 3, 5, and 7 (biweekly); Patient 6 (never).

## Data Availability

Data are contained within the article and Appendix A.

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
