# Peer review of "Long-Term Efficacy and Safety of Evinacumab in Patients with Homozygous Familial Hypercholesterolemia: Real-World Clinical Experience"

_pharmaceuticals, 2022, doi:10.3390/ph15111389_

Round 1
Reviewer 1 Report
The paper by Stefanutti et al., is well-written and original. This is the first report about evinacumab treatment in HoFH patients in a real-world setting and provided confirmatory findings about the effectiveness of this therapeutic approach in severe hypercholesterolemia.
Minor comments:
- On which basis lipid apheresis was stop, when starting compassionate use?
- Patient 1 had an LDL-C rebound after switching to compassionate use. Please discuss possible reason if available (stop of LA?)
- Are ASCVD surrogate markers available to assess the efficacy of evinacumab on CVD?
Author Response
Response to Reviewer 1
Thank you very much for your very kind comments on the originality of our paper. In fact, we thought that they were the first real world data available on the efficacy and safety of evinacumab and that it was important to bring them to the attention of the international scientific community.
Q1: On which basis lipid apheresis was stop, when starting compassionate use?
Response: We acknowledge the reviewer’s comment. During the 24 months of evinacumab intervention, it was not possible to change the lipid-lowering therapy (LLT), including lipoprotein apheresis (LA). Once the 24-month follow-up was concluded, it became clear that by interrupting the extracorporeal treatment and leaving the pharmacological treatment plus evinacumab in place, the lipid profile did not undergo any significant change except for patient #4 in which the elevated concentration of Lp(a) persisted despite showing exceptional results of the lipid profile. All patients were then transitioned into a compassionate program to continue on evinacumab at the same dose without LA treatment.
Q2: Patient 1 had an LDL-C rebound after switching to compassionate use. Please discuss possible reason if available (stop of LA?)
Response: We acknowledge the reviewer’s comment. Patient 1 interrupted therapy for hospitalization due to a condition unrelated to her severe hypercholesterolaemia; LLT was re-instituted following her recovery, with subsequently marked reduction in LDL-C. Accordingly, we have modified the text to clarify the issue.
Line 93-95 “The patient discontinued LLT for hospitalization due to a condition unrelated to her severe hypercholesterolaemia; LLT was re-instituted following recovery, with subsequently marked and sustained reduction in LDL-C.”
Q3: Are ASCVD surrogate markers available to assess the efficacy of evinacumab on CVD?
Response: We thank the reviewer for this suggestion. As reported in the manuscript, serial hsCRP values were determined during the 24 months of evinacumab intervention (Supplementary material Figure S3); hsCRP concentration decreased impressively in 2 patients increased in 2 patients. During the compassionate use phase, hsCRP was not determined. Hence, it is difficult to draw a conclusion on the efficacy of evinacumab on ASCVD based on hsCRP data. However, there was no cardiovascular event observed during the 24-month follow-up and subsequent compassionate extension period with evinacumab. Furthermore, carotid intima-media thickness (IMT) was also measured as a surrogate marker for atherosclerotic cardiovascular disease (ASCVD) in our entire cohort of HoFH patients, whether or not treated with evinacumab. However, the data have not yet been analyzed.
Accordingly, we have addressed this issue in the revised manuscript.
Line 148 “There was no cardiovascular event observed during the 24-month follow-up and subsequent compassionate extension period (12 months) with evinacumab.”
Lines 230-233 “Carotid intima-media thickness (IMT) has also been used as a surrogate marker for ASCVD in FH patients. Whether evinacumab therapy has favourable effect on the progression of carotid IMT is under investigation in our HoFH patients.”
Reviewer 2 Report
This a real-life study on the effect of evinacumab on plasma lipid levels in patients with HoFH (n=7).
Comments
1. It is not clear to me when patients stopped LA. An increase in LDL-C would be expected after stopping LA, but this is not the case in the manuscript.
2. Which LDL-C did authors measured? Pre-LA, post-LA or mean LDL-C?
3. Lomitapide use should be discussed in the manuscript. Why only one patient was on lomitapide? Any lomitapide use in the past in the other patients?
4. Some grammatical issues should be corrected: a) line 48: needed, b) Table 1; Patient 3: HoFH, c) line 164: of LDL-C, d) line 177: performed (instead of investigated), e) line 195: omit ‘which’, f) line 237: require, g) line 238: Given that.
5. Line 91: plasma concentration of??
Author Response
Response to Reviewer 2
Thank you for your comment. Indeed, this was what we wished to bring to the attention of the community of international specialists.
Q1: It is not clear to me when patients stopped LA. An increase in LDL-C would be expected after stopping LA, but this is not the case in the manuscript.
Response: We acknowledge the reviewer’s comments. The rebound of LDL-C was due to the interruption of the LA to allow the correct evaluation of the impact of the drug treatment alone before entering the compassionate use phase. We would like to mention that the patients at the time of the discontinuation of LA were still on fully effective treatment with evinacumab and concomitant background LLT. Also see response to Reviewer 1 Q1.
Q2: Which LDL-C did authors measured? Pre-LA, post-LA or mean LDL-C?
Response: We acknowledge the reviewer’s comments. We reported LDL-C as mean LDL-C
Q3: Lomitapide use should be discussed in the manuscript. Why only one patient was on lomitapide? Any lomitapide use in the past in the other patients?
Response: We thank the reviewer for this suggestion. The multinational group of the phase III clinical trial investigators (including our group) published their results on Lancet and got the EMA approval not only for clinical use in Italy but for Europe and US. Currently, 4 of our FH patients are subjected to lomitapide with variable doses, but not exceeding 30 mg /day. Only one patient of the 'evinacumab' subcohort continued lomitapide treatment with six-monthly liver ultrasound and annual liver MRI, in addition to biochemical tests of liver function. The patient tolerates the drug well for the prescribed dose. In our opinion, evinacumab is superior to lomitapide with less side effects deriving from the dosage increase of lomitapide for which some of our patients have asked to stop the relative prescription. Accordingly, we have addressed this issue in the revised manuscript.
Lines 239-243 “Lomitapide is approved by European Medicines Agency (EMA) for clinical use in Italy. Currently, 4 of our FH patients are subjected to lomitapide with variable doses, but not exceeding 30 mg /day. Only one patient of the 'evinacumab' subcohort continued lomitapide treatment. In our experience, evinacumab is superior to lomitapide with less side effects deriving from the dosage increase of lomitapide.”
Q4: Some grammatical issues should be corrected: a) line 48: needed, b) Table 1; Patient 3: HoFH, c) line 164: of LDL-C, d) line 177: performed (instead of investigated), e) line 195: omit 'which', f) line 237: require, g) line 238: Given that.
Response: We thank the reviewer for alerting the grammatical mistakes. Accordingly, we have corrected the grammatical mistakes in the revised manuscript
Line 48 “New therapies are needed to target LDL metabolism beyond the LDL receptor pathway in patients with HoFH, to be used alone or in combination with LA [1, 7, 8].
Table 1 “True HoFH”
Line 178 “More importantly, these studies were not performed in a real-world clinical setting.”
Line 195 “Direct uptake by the liver of VLDL remnants, together with the fall in the hepatic output of VLDL may contribute to reduction in the production of LDL particles and LDL-C concentration [15].”
Line 246 “However, treatment responses to statins and PCSK9 inhibitors in HoFH patients are suboptimal because these agents require residual LDL receptor function [1, 8].”
Line 247 “Given that the mechanism of action of ANGPTL3 inhibitors appears to be independent of LDL receptor activity [18, 22],”
Q5: Line 91: plasma concentration of??
Response: We thank the reviewer for alerting the mistake.
Line 91 “in this patient the corresponding on-treatment plasma concentration of LDL-C at 24-month follow-up returned approximately to pre-treatment levels”